Description of two new Apseudopsis species (A. larnacensis sp. nov and A. salinus sp. nov.) (Tanaidacea: Crustacea) from the Mediterranean and a biogeographic overview of the genus

Stępień Anna 1 anna.stepien@biol.uni.lodz.pl
http://orcid.org/0000-0002-5171-3145 Jóźwiak Piotr 1
Gómez Sergio C. Garcia 2
Avramidi Eleni 3
Grammatiki Kleopatra 3
Lymperaki Myrsini 4
Küpper Frithjof C. 3 5 6
Esquete Patricia 7
1 Faculty of Biology and Environmental Protection, University of Lodz , Lodz , Poland
2 Tecnoambiente S.L. , Jerez de la Frontera , Spain
3 School of Biological Sciences, University of Aberdeen , Aberdeen, Scotland , United Kingdom
4 Centro de Ciências do Mar, Universidade do Algarve , Faro , Portugal
5 Marine Biodiscovery Centre, Department of Chemistry, University of Aberdeen , Aberdeen, Scotland , United Kingdom
6 Department of Biology, San Diego State University , San Diego, California , United States
7 Departamento de Biologia & Centro de Estudos do Ambiente e do Mar, Universidade de Aveiro , Aveiro , Portugal
Yapıcı Sercan
Electronic publication date: 2024 Dec 24
Publication date: 2024
Volume: 12
Electronic Location ID: e18740
Received 2024 Sep 23; Accepted 2024 Nov 29
Copyright: © 2024 Stępień et al.
Copyright year: 2024
Copyright holder: Stępień et al.
License: This is an open access article distributed under the terms of the Creative Commons Attribution License, which permits unrestricted use, distribution, reproduction and adaptation in any medium and for any purpose provided that it is properly attributed. For attribution, the original author(s), title, publication source (PeerJ) and either DOI or URL of the article must be cited.
License URL: https://creativecommons.org/licenses/by/4.0/

Keywords: Benthos, Apseudidae, Ecoregions, Peracarida

Funding: H2020 European project WATER-MINING 869474 Marine Alliance for Science and Technology for Scotland pooling initiative Scottish Funding Council HR09011 National Funds (OE) FCT–Fundação para a Ciência e a Tecnologia, I.P Decree-Law 57/2016 of 29 August, changed by Law 57/2017 of 19 July CESAM by FCT/MCTES (UIDP/50017/2020 + UIDB/ 50017/2020 + LA/P/0094/2020) University of Lodz This work was supported by the European Commission for supporting the activities carried out in the framework of the H2020 European project WATER-MINING (project under grant agreement No. 869474). This work also received support from the Marine Alliance for Science and Technology for Scotland pooling initiative. MASTS is funded by the Scottish Funding Council (grant reference HR09011) and contributing institutions. Patricia Esquete is funded by national funds (OE), through FCT—Fundação para a Ciência e a Tecnologia, I.P., in the scope of the framework contract foreseen in the numbers 4, 5 and 6 of article 23 of the Decree-Law 57/2016 of 29 August, changed by Law 57/2017 of 19 July. Financial support was received from CESAM by FCT/MCTES (UIDP/50017/2020 + UIDB/ 50017/2020 + LA/P/0094/2020), through national funds. The APC was supported by the University of Lodz. The funders had no role in study design, data collection and analysis, decision to publish, or preparation of the manuscript.

==============================
The Mediterranean Sea is recognized as one of the most threatened marine environments due to pollution, the unintentional spread of invasive species, and habitat destruction. Understanding the biodiversity patterns within this sea is crucial for effective resource management and conservation planning. During a research cruise aimed at assessing biodiversity near desalination plants in the vicinity of Larnaca, Cyprus, conducted as part of the WATER-MINING project (Horizon 2020), specimens of the tanaidacean genus Apseudopsis were collected. These were classified into two species, identified as new to science, and are described herein as Apseudopsis larnacensis sp. nov and Apseudopsis salinus sp. nov. Apseudopsis larnacensis sp. nov is distinguished from its congeners by the presence of a lateral apophysis on pleonite 5, which is longer than those on pleonites 1–4, hyposphenia on pereonites 2–6, and six ventral spines on the propodus of pereopod 1. Apseudopsis salinus sp. nov. is characterized by a rounded posterolateral margin on pereonite 6, hyposphenia on pereonites 1–6, and four ventral spines on the propodus of pereopod 1. An updated key to the Atlantic and Mediterranean Apseudopsis species is provided. This study provides an overview of the distribution, bathymetric, and habitat preference of all known Apseudopsis species. Data extracted from the literature and two popular online databases were analyzed to identify diversity patterns across seas and ecoregions. Our analysis reveals that the Mediterranean is the most diverse region for Apseudopsis, with the Levantine Sea hosting ten species and the South European Atlantic Shelf seven species. Shallow waters and muddy and sandy habitats are conducive for Apseudopsis occurrence.

Introduction

The Mediterranean Sea is recognized as a marine biodiversity hotspot, harboring between 4% and 18% of the world’s marine species richness (Bianchi & Morri, 2000; Arvanitidis et al., 2002; Coll et al., 2010). Despite extensive studies conducted since ancient times, our understanding of the region’s biodiversity remains incomplete, with new species still being discovered (Sturaro & Guerra-García, 2012; Beli et al., 2018; Gómez et al., 2024). Accurately identifying biodiversity patterns in the Mediterranean is a crucial challenge, particularly considering pressing environmental issues such as climate change, pollution, and biological invasions (Coll et al., 2012; Deudero & Alomar, 2015; Kletou, 2019).

Given the Mediterranean’s importance as biodiversity hotspot (Bianchi & Morri, 2000; Coll et al., 2010, 2012; Myers et al., 2000) and the ongoing challenges posed by human activities and environmental changes, one of the objectives of the ‘WATER-MINING’ project within the framework of Horizon 2020 aimed to enhance our understanding of the impacts of seawater desalination plants on marine biodiversity, by studying the condition of Posidonia oceanica seagrass meadows near the hypersaline outfalls of two major desalination plants in Cyprus, Larnaca and Dhekelia (Xevgenos et al., 2021). During the investigation, specimens of the genus Apseudopsis Norman, 1899 were collected. Apseudopsis belongs to the family Apseudidae within the crustacean order Tanaidacea. Identification of its representatives is often hindered by the high number of sympatric species and intricate history of taxonomic studies (Lang, 1955; Bamber et al., 2009; Esquete, Fersi & Dauvin, 2019).

Investigations of the diversity of Apseudopsis date back to the 19th century, when the genus was established to distinguish species of the family Apseudidae Leach, 1814, with eye lobes fused to the carapace (later classified under several new genera) from species with a clear demarcation (classified as Apseudes Leach, 1814) (Norman, 1899). Two species were initially included in this new genus: Apseudopsis acutifrons (Sars, 1882), and Apseudopsis hastifrons (Norman & Stebbing, 1886). Additional species belonging to the genus Apseudopsis were not identified until 48 years later, when A. ostroumovi Băcescu & Carausu, 1947, was discovered and described from the Black Sea. The prolonged gap between the establishment of the genus and the identification of further representatives may have been due to the unclear definition of Apseudopsis (Lang, 1949, 1955).

To address the difficulties in distinguishing Apseudes from Apseudopsis, Lang reexamined specimens of A. acutifrons collected from the vicinity of Naples, specimens of A. hastifrons from the Adriatic and Atlantic coasts of Morocco, and specimens of A. ostroumovi from the Black Sea. His investigation resulted in the invalidation of the genus Apseudopsis and its synonymization with Apseudes (Lang, 1955). Additionally, Lang synonymized Apseudes ostroumovi and A. hastifrons with A. acutifrons (Lang, 1955).

Nearly fifty years later, an examination of a large collection of Apseudidae from the Mediterranean and the North African Atlantic enabled Guţu (2002) to revise certain species of Apseudes. As a result, several new species were established that would later be included in Apseudopsis. These included Apseudes bacescui Guţu, 2002, from the Spanish coast, previously identified as A. ostroumovi, and Apseudes arguinensis Guţu, 2002, from the Atlantic coast of Mauritania, previously recognized as A. hastifrons, Apseudopsis annabensis (Guţu, 2002) from Mauritania and Apseudopsis apocryphus (Guţu, 2002), Apseudopsis minimus (Guţu, 2002) and Apseudopsis tridens (Guţu, 2002) from the coast of Israel. Further studies of this and other collections allowed Guţu (2006) to revalidate the genus Apseudopsis.

He reinstated A. acutifrons as the type species of the reestablished genus, and transferred two species: Apseudes latreillii, described by Milne Edwards, 1828 from the Atlantic coast of France, and A. uncidigitatus Norman & Stebbing, 1886 from the Mediterranean coast of Africa.

Finally, a revision of several populations from different parts of the Mediterranean enabled Guţu to revalidate the species Apseudopsis ostroumovi, A. hastifrons, and A. latreillii mediterraneus, all of which had previously been synonymized with A. latreillii (Lang, 1955; Sieg, 1983). Guţu also elevated A. latreillii mediterraneus to full species rank (Guţu, 2002, 2006). In the process of reestablishing the genus Apseudopsis several species were transferred to the genus: A. elisae (Băcescu, 1961), A. erythraeicus (Băcescu, 1984) from Aqaba Bay (Red Sea), A. olimpiae (Guţu, 1986) from Bermuda, and A. bruneinigma (Bamber, 1999) from the coast of Brunei. Additionally, species described by Guţu himself in 2002, such as A. bacescui, A. arguinensis, A. annabensis from the Algerian coast, and A. apocryphus, A. minimus, and A. tridens from the Israeli coast, were also included. A. caribbeanus Guţu, 2006, from the Gulf of Batabanó in Cuba, and A. isochelatus Guţu, 2006, from the coast of Mauritania were later described.

The knowledge about the distribution of Apseudopsis was expanded with the discovery of A. tuski (Błażewicz-Paszkowycz & Bamber, 2007) from Bass Strait, Australia, which was initially recognized as Apseudes and later transferred to Apseudopsis (Błażewicz-Paszkowycz & Bamber, 2012). The presence of Apseudopsis in Brunei and Atlantic waters was further confirmed by the discovery of two new species: A. opisthoscolops Bamber, Chatterjee & Marshall, 2012, in Brunei, and A. cuanzanus Bochert, 2012, in Atlantic waters.

Investigations conducted over the last decade have provided significant insights into intraspecific variation, taxonomic characteristics, and the distribution of Apseudopsis in the Mediterranean and surrounding Atlantic coasts (Esquete et al., 2012a, 2012b; Esquete, Ramos & Riera, 2016; Esquete, Fersi & Dauvin, 2019; Carvalho et al., 2019; Senckenberg Ocean Species Alliance et al., 2024). Alongside the description of A. adami from the Iberian Peninsula, Esquete et al. (2012a) also documented its life history and developmental stages, and listed a series of characters that remain stable throughout all life like the shape of the rostrum, the presence and/or position of apophyses on pereonites, the number of hyposphenia on neuters and adult females without marsupium, and the number of spines on pereopod 1 propodus (Esquete et al., 2012a, 2012b). A key to the genus, as well as the description of A. rogi from Tenerife, were detailed in a article by Esquete, Ramos & Riera (2016). Finally, two new species, A. formosus from Portugal and A. gabesi from Tunisia, along with new records of already known species, were presented by Esquete, Fersi & Dauvin (2019) and Carvalho et al. (2019). Diversity of Apseudopsis off the Mediterranean coast of Israel with summary of the knowledge about those species was presented by Lubinevsky, Tom & Bird (2022). Most recently, A. daria Esquete & Tato, 2024 was discovered from Ria de Ferrol (Galicia: Spain) (Senckenberg Ocean Species Alliance et al., 2024).

Currently 26 nominal species of Apseudopsis are recognized. Recent studies in the Ria Formosa (east of Faro, Algarve, Portugal) or the Gulf of Gabes (Tunisia), where six and four species were discovered, respectively—one of which was new to science in each area (Carvalho et al., 2019; Esquete, Fersi & Dauvin, 2019)—indicate that the actual diversity might be much higher.

The aim of our article is to describe two new members of Apseudopsis that were found in the Cyprus Basin, in the vicinity of a desalination plant in Larnaca. We also update the identification key to the known Atlantic and Mediterranean Apseudopsis species and summarize the existing knowledge about the ecology and biogeography of the genus based upon literature, the Ocean Biodiversity Information System (OBIS) and the Global Biodiversity Information Facility (GBIF). With this synoptic basis, this study explores the larger biodiversity patterns of the genus Apseudopsis.

Materials and Methods

Study area

The samples were collected in Larnaca Bay, within the Cyprus Basin and larger Levantine Basin located in the eastern part of the Mediterranean. The area is classified as the Levantine Ecoregion according to Spalding et al. (2007). The Levantine Basin is characterized by warm, salty, and oligotrophic waters. In general, the surface water reaches nearly 39 PSU and temperature ranges between 15–30 °C, depending on season (Zodiatis, Theodorou & Demetropoulos, 1998; Habib et al., 2022). The Cyprus Basin is under influence of the Asia Minor Current flowing in a north–westerly direction and the mid-Mediterranean jet moving towards the south-western part of the Levantine Basin (Zodiatis et al., 2005).

Sampling

The material was collected during inshore 1-day cruises using the vessels of the Department of Fisheries and Marine Research of the Cyprus Government, organized within the framework of the EU Horizon 2020 project ‘WATER-MINING’ in 2022. Benthos samples were gathered using a Van Veen grab with a surface area 0.05 m2. On board samples were sieved through a 0.5 mm mesh, stained with rose bengal and preserved in formaldehyde (4%) or ethanol (96%). A total of 16 specimens, classified to two species of the genus Apseudopsis, were collected on station with coordinates 34°52′02.6″N, 33°39′14.0″E, from a depth of about 11 m, from muddy sand sediments.

Morphological examination and taxonomic description

Selected individual of each morphospecies were dissected with use of chemically sharpened needles. Appendages were placed on drop of glycerin on microscope slides, protected with cover glass and sealed with melted paraffin. Slides were used for drawings, which were prepared with the use of a light microscope (Nikon Eclipse 50i) equipped with camera lucida. Hight quality illustrations were drawn using digital tablet and Adobe Illustrator software (Coleman, 2003).

Type materials were deposited at Senckenberg Natural History Museum at Frankfurt.

The electronic version of this article in Portable Document Format (PDF) will represent a published work according to the International Commission on Zoological Nomenclature (ICZN), and hence the new names contained in the electronic version are effectively published under that Code from the electronic edition alone. This published work and the nomenclatural acts it contains have been registered in ZooBank, the online registration system for the ICZN. The ZooBank LSIDs (Life Science Identifiers) can be resolved and the associated information viewed through any standard web browser by appending the LSID to the prefix http://zoobank.org/. The LSID for this publication is: (urn:lsid:zoobank.org:pub:72BC0082-9C17-4901-8D9A-547B073760EB). The online version of this work is archived and available from the following digital repositories: PeerJ, PubMed Central SCIE and CLOCKSS.

Biogeography, bathymetry and habitat preferences data

Occurrence data for the members of Apseudopsis were extracted from relevant literature (based on literature set by Anderson (2023)). The geographic distribution data were supplemented with information downloaded from two online databases: Ocean Biodiversity Information System OBIS (obis.org) and Global Biodiversity Information Facility GBIF (gbif.org). This online database was the source of 1,579 records for ten species. Four hundred eighty-seven coordinates were collected from literature, that were given for 25 species and were related mostly to the type locality. To ensure comprehensive coverage of the distribution of Apseudopsis members, we have calculated coordinates for species, whose localities were given as name of areas without explicit coordinates. These coordinates were calculated with Google Maps considering only precise locality descriptions e.g., city name/port/bay. All data with coordinates were merged and checked for reliability with use of quality control tools being part of Robis package in R (Provoost & Bosch, 2020). Duplicates, records on land and records with a distance uncertainty higher than 100 km were removed from the matrix. Moreover, the matrix was checked for relevance of scientific name of taxa, if needed synonyms were updated. In total 2,081 records of Apseudopsis were found (including coordinates for two species provided in the current article; Table S1). The matrix was used for further analysis. The number of species was calculated for larger regions, such as Mediterranean or Black Sea, and in detail for ecoregions (Spalding et al., 2007). Furthermore, the number of endemic species and number of records were related to ecoregions. Data plotting and manipulating were performed with following R packages: “tidyverse” (Wickham et al., 2019), “ggplot2” (Wickham, 2016), “sf” (Pebesma, 2018). A distribution matrix used for map was prepareded using QGIS 3.32.2.

Bathymetric information and habitat preferences of Apseudopsis members were extracted from literature and from online databases. Depth information was given for 23 species (and for two described herein), information for A. arguinensis, A. erythraeicus, and A. tridens were not given. Habitat preference data was extracted for 20 species, while preferences of the remaining eight (A. arguinensis, A. cuanzanus, A. erythraeicus, A. hastifrons, A. isochelatus, A. olimpiae, A. tridens, A. tuski) were not provided.

Results

Taxonomy

Genus Apseudopsis Norman, 1899

Type species: A. acutifrons (Sars, 1882)

Species included: A. acutifrons (Sars, 1882), A. adami Esquete & Bamber, 2012, A. annabensis (Guţu, 2002), A. apocryphus (Guţu, 2002), A. arguinensis (Guţu, 2002), A. bacescui (Guţu, 2002), A. bruneinigma (Bamber, 1999), A. caribbeanus Guţu, 2006, A. cuanzanus Bochert, 2012, A. daria Esquete & Tato, 2024, A. elisae (Băcescu, 1961), A. erythraeicus (Băcescu, 1984), A. formosus Carvalho, Pereira & Esquete, 2019, A. gabesi Esquete, 2019, A. hastifrons (Norman & Stebbing, 1886), A. isochelatus Guţu, 2006, A. latreillii (Milne Edwards, 1828), A. mediterraneus (Băcescu, 1961), A. minimus (Guţu, 2002), A. olimpiae (Guţu, 1986), A. opisthoscolops Bamber, Chatterjee & Marshall, 2012, A. ostroumovi Băcescu & Carausu, 1947, A. rogi Esquete, 2016, A. tridens (Guţu, 2002), A. tuski (Błażewicz-Paszkowycz & Bamber, 2007), A. uncidigitatus (Norman & Stebbing, 1886)

Apseudopsis larnacensis Esquete, Stępień, Jóźwiak sp. nov.

urn:lsid:zoobank.org:pub:72BC0082-9C17-4901-8D9A-547B073760EB

(Figs. 1–3)

Figure 1 Apseudopsis larnacensis sp. nov., holotype, preparatory female (SMF 57060).

Habitus illustration. (A) Dorsal view. (B) Lateral view. Arrows indicate diagnostic features. Scale bar = 1 mm. Photo credit: Anna Stępień.

Figure 2 Apseudopsis larnacensis sp. nov., paratype, preparatory female (SMF 57061).

Antenna, antennule and mouth parts illustration. (A) Antennule. (B) Antenna. (C) Labrum. (D) Left mandible. (D′) Molar of left mandible. (D′′) Palp of left mandible. (E) Right mandible. (F) Maxillule. (F′) Palp of maxillule. (G) Maxilla. (H) Labium. (I) Maxilliped. (J) Epignath. Scale bar = 0.1 mm.

Figure 3 Apseudopsis larnacensis sp. nov., paratype, preparatory female (SMF 57061).

Cheliped and pereopods illustration. (A) Cheliped outer side. (A′) Chela inner side. (B) Pereopod 1. (C) Pereopod 2. (D) Pereopod 3. (E) Pereopod 4. (F) Pereopod 5. (G) Pereopod 6. (H) Pleopod 1. (I) Uropod. Scale bar = 0.1 mm.

Material: Holotype: preparatory female (SMF 57060), vicinity of Larnaca, Cyprus, 34°52′02.6″N, 33°39′14.0″E, depth about 11 m, muddy sand; paratypes: nine preparatory females (SMF 57061) (two dissected), the same station as holotype.

Etymology: Named after the city Larnaca, in the vicinity of which the species was found.

Diagnosis: prominent anterolateral and posterolateral apophyses on pereonites 2–5, hyposphenia present on pereonites 2–6, lateral apophyses on pleonite 5 two times longer than apophysis on pleonites 1–4, six ventral spines on pereopod 1 propodus.

Description of preparatory female

Total length 8.0–8.5 mm. Body (Figs. 1A, 1B) slightly narrowed posteriorly, 5.8 times as long as wide. Cephalothorax about as long as wide, about 1/6 of total body length; rostrum acute, almost trifid, similar to A. ostromouvi described by Băcescu & Carausu. Ocular lobes present, pointed. Pereonites with simple lateral setae, pereonites 2–6 with hyposphenia. Pereonite 1 posterolateral corners with curved apophyses. Pereonites 1 to 5 subrectangular. Pereonite 6 with anterolateral acute apophyses. Pleonites subequal, lateral margins of all pleonites, with ventrally oriented pointed apophyses. Pleonite 5 slightly longer. Pleotelson at least 3.3 times as long as wide, lateral setae and one pair of subterminal long setae.

Antennule (Fig. 2A) peduncle first article 3.2 times as long as wide, outer margin with one simple and one penicillate mid-length setae and subdistal tuft of simple and penicillate setae, inner margin with simple setae. Second article about half length of first, twice as long as wide, with inner and outer subdistal tufts of simple setae and four subdistal penicillate setae. Third article half-length of second, with inner and outer simple setae. Fourth (common) article as long as third, naked. Main flagellum of ten segments, with five aesthethascs and simple setae. Accessory flagellum of three segments, with simple setae.

Antenna (Fig. 2B) peduncle first article as long as wide, with inner lobe with small seta. Second article 1.5 times as long as first, 1.5 times as long as wide, with one inner and one outer small setae and four subdistal setae, bearing an outer squama with long marginal setae. Third article shorter than wide, naked. Fourth article about twice as long as third, naked. Fifth article 1.3 times as long as two preceding articles together, with simple setae and penicillate setae. Flagellum of seven segments, first segment with long simple setae on outer margin, rest with simple and penicillate setae.

Mouthparts. Labrum bilobed, setulose (Fig. 2C). Left mandible (Figs. 2D, 2D′) with strongly dentate pars incisiva, lacinia mobilis tridentate; setiferous lobe with five bifurcate and trifurcate setae; pars molaris triturative. Mandibular palp (Fig. 2D′′) three-articled; first article about twice as long as wide, with numerous setae on inner margin; second article 1.3 times as long as first, with 11 pectinate short setae and ten simple setae; distal article with five pectinate setae and several simple setae. Right mandible (Fig. 2E) as left but without lacinia mobilis. Maxillule (Fig. 2F) inner endite with four distal pectinate setae and one subdistal simple seta; outer endite with ten distal spines and two subdistal setae, outer margin with fine setae; palp (Fig. 2F′) with three subterminal and two terminal minutely plumose setae. Maxilla (Fig. 2G) with inner margin serrate, outer lobe of fixed endite with bifurcate, trifurcate and pectinate spines; outer lobe of fixed endite with a row of numerous simple setae in front of four pectinate setae; inner lobe of moveable endite with serrate setae. Labium (Fig. 2H) with outer margin serrated, palp with three distal spines and marginal setules. Maxilliped (Fig. 2I) basis outer distal corner with serrate lobe with setules; palp first article with one outer distal seta; second article inner margin with two rows of numerous setae, outer distal margin with one long distal seta; third article with two rows of setae along inner margin; distal article with seven distal setae, four setae stronger than rest; endite with subdistal setules on outer margin; inner margin with five coupling-hooks, vental margin with row of 11 setulose spines, distal margin with simple and blunt setae. Epignath (Fig. 2J) with long, distally setulose spine.

Cheliped slender (Figs. 3A, 3A′). Basis subtriangular, 1.9 times as long as wide, with mid-ventral stout spine, mid-ventral tuft of simple setae and simple setae on ventral and dorsal margins. Exopodite present; first article cylindrical, naked, distal article with four or five plumose setae. Merus elongate, narrower proximally, with medial and ventrodistal groups of simple setae. Carpus slender, 2.5 times as long as wide, with distal projection, with simple setae laterally and on ventral margin. Chela twice as long as wide; palm about as long as wide, with dorsal and lateral simple setae, and one pectinate spine near dactylus insertion (Fig. 3A′); fixed finger with row of ventrodistal setae; cutting edge almost straight, with row of setae. Dactylus as long as fixed finger, cutting edge with small spines.

Pereopod 1 (Fig. 3B) coxa with pronounced anterodistal apophysis. Basis cylindrical, twice as long as wide, with simple setae along anterior and posterior margins; exopodite as on cheliped. Ischium with two simple ventrodistal setae. Merus narrower proximally, 1.8 times as long as wide, with simple setae, and ventrodistal spine. Carpus 0.7 times as long as merus, with marginal setae, diagonal row of setae, two ventral spines and one dorsodistal spine. Propodus about as long as carpus, with marginal setae, and two dorsodistal spines. Dactylus with three fine setae, one near unguis and one mid-dorsal simple seta. Unguis about one-quarter of length of dactylus.

Pereopod 2 (Fig. 3C) basis cylindrical, 2.2 times as long as wide, with dorsal and ventral simple setae, and tuft of ventrodistal setae. Ischium with ventrodistal setae. Merus, carpus and propodus with long marginal setae. Merus narrower at base, 1.4 times as long as wide, with one long ventrodistal spine and two medial, shorter spines. Carpus narrower at base, about as long as merus, with simple setae along ventral and dorsal margins, and shorter setae laterally. Propodus 2.7 times as long as wide, with one ventral, slender spine. Dactylus 0.6 times as long as propodus, unguis about half of length of dactylus.

Pereopod 3 (Fig. 3D) similar to pereopod 2, but basis 3.0 times as long as wide, with simple setae and row of three penicillate setae and ventrodistal tuft of simple setae. Ischium with ventral setae. Merus, carpus and propodus with long marginal setae. Merus narrower at base, 1.4 times as long as wide, with row of medial long setae. Carpus about as long as merus, narrower at base, 1.4 times as long as wide, with row of medial long setae.

Pereopod 4 (Fig. 3E) basis oval, 2.4 times as long as wide, with tuft of ventrodistal setae. Ischium with ventrodistal setae. Merus, carpus and propodus with long ventral and lateral setae. Merus narrower at base, about as long as wide, with one ventrodistal spine and ventrolateral spine. Carpus 1.5 times as long as wide, with five long ventral spines and three lateral spines. Propodus slender, 2.5 times as long as wide and about as long as carpus, with one dorsoproximal penicillate seta, tuft of distal setae, row of unequal pectinate setae and two ventral slender spines. Dactylus 0.7 times as long as propodus, with one short setule near unguis. Unguis about one-third of length of dactylus.

Pereopod 5 (Fig. 3F) basis oval, 2.4 times as long as wide, with tuft of ventrodistal setae and dorsal penicillate setae. Ischium with ventrodistal setae. Merus, carpus and propodus with long ventral and lateral setae. Merus narrower at base, about as long as wide. Carpus about twice as long as wide, with setae in two rows. Propodus about as long as carpus, with dorsoproximal penicillate seta, subdistal pectinate spines and one distally setulose spine. Dactylus 0.5 times as long as propodus, with distal setule. Unguis about one-third of length of dactylus.

Pereopod 6 (Fig. 3G) Basis oval, twice as long as wide, with row of dorsal and ventral plumose setae. Ischium with a ventrodistal tuft of simple setae. Merus and carpus with ventral simple setae. Merus narrower proximally, with rows of ventral simple and long plumose dorsal setae, and three short medial setae. Carpus with four ventral spines, longer towards propodus, row of long plumose dorsal setae and row of short medial setae. Propodus ovate, 0.8 times as long as carpus, with row of simple medial setae and row of pinnate spines along ventral and terminal border. Dactylus about as long as propodus, with one medial and one distal dorsal setules. Unguis about one-third of length of dactylus.

Pleopods 1 to 5 respectively with eight outer and 11 inner plumose setae (Fig. 3H), eight outer and nine inner, seven outer and ten inner, nine outer and seven inner and seven outer and five inner plumose setae. Both rami with numerous distal and outer marginal plumose setae, endopod with inner marginal seta.

Uropod (Fig. 3I) protopod with two inner-distal setae. Endopod with 25 to 30 segments, some with simple or penicillate setae. Exopod three-segmented; distal segment significantly longer than the other two together, with two long distal setae.

Distribution: Species known only from the type locality.

Remarks

This species can be distinguished from any other species of the genus because of the uniquely long lateral apophyses on pleonite 5. Otherwise, is most similar to Apseudopsis hastifrons (Guţu, 2002) from the Gulf of Naples in having six ventral spines on pereopod 1 propodus and anterolateral and posterolateral apophyses on pereonites 2–5, but it differs in having no posterolateral apophyses on pereonite 6, hyposphenia on pereonites 1–6 (only on pereonites 2, 3–4 and 6 on A. hastifrons) and in lacking a crown of proximal spines on the pereonite 5 basis, the latter indicated by Guţu (2002) as definitive of the species.

Apseudopsis salinus Esquete, Jóźwiak, Stępień, sp. nov.

urn:lsid:zoobank.org:pub:72BC0082-9C17-4901-8D9A-547B073760EB

(Figs. 4–6)

Figure 4 Apseudopsis salinus sp. nov., holotype, preparatory female (SMF 57062).

Habitus illustration. (A) Dorsal view. (B) Lateral view. Arrows indicate diagnostic features. Scale bar = 1 mm. Photo credit: Anna Stępień.

Figure 5 Apseudopsis salinus sp. nov., paratype, preparatory female (SMF 57063).

Antenna, antennule and mouth parts illustration. (A) Antennule. (B) Antenna. (C) Labrum. (D) Left mandible. (E) Right mandible. (F) Maxillule. (G) Maxilla. (H) Labium. (I) Maxilliped. (I′) Maxilliped endite. Scale bar = 0.1 mm.

Figure 6 Apseudopsis salinus sp. nov., paratype, preparatory female (SMF 57063).

Cheliped and pereopods illustration. (A) Cheliped, outer side. (A′) Chela inner side. (B) Pereopod 1. (C) Pereopod 2. (D) Pereopod 3. (E) Pereopod 4. (F) Pereopod 5. (G) Pereopod 6. (H) Pleopod 1. (I) Uropod. Scale bar = 0.1 mm.

Material: Holotype: preparatory female (SMF 57062), vicinity of Larnaca, Cyprus, 34°52′02.6″N, 33°39′14.0″E, from a depth of about 11 m, muddy sand; paratypes: five preparatory females (SMF 57063) (two dissected), the same station as holotype.

Etymology: The species occurs in the vicinity of a brine discharge point from a desalination plant in Larnaca, and therefore the salinity values are rather high. “Salinus” means “saline”, or “containing salt” in Latin.

Diagnosis: prominent anterolateral and posterolateral apophyses on pereonites 2–5; hyposphenia present on pereonites 1–6, apophyses on pleonites 1–5 similar in length, four ventral spines on pereopod 1 propodus.

Description of preparatory female

Total length 6.1–6.3 mm. Body (Figs. 4A, 4B) slightly narrowed posteriorly, six times as long as wide. Cephalothorax about as long as wide, about 1/6 of total body length; rostrum acute, downturned, with rounded “shoulders”. Ocular lobes present, long, acute and directed forward. Pereonites with simple lateral setae. Pereonite 1 posterolateral corners rounded. Pereonite 2 with posterolateral pointed apophyses, pereonite 3–5 with medial apophyses directed forward and posterolateral apophyses directed backwards, pereonite 6 with medial apophyses directed forward. Pleonites subequal, lateral margins of all pleonites produced posteriorly, with ventrally oriented pointed apophyses. Pleotelson about three times as long as wide, with one pair of setae and one pair of subterminal long setae.

Antennule (Fig. 5A) peduncle first article three times as long as wide, outer margin with tuft of proximal penicillate setae, tuft of penicillate and one medial simple setae and subdistal tuft of simple and penicillate setae, inner margin with simple setae. Second article about half length of first, twice as long as wide, with subdistal simple setae and penicillate setae. Third article half-length of second, with inner and outer simple setae. Fourth (common) article as long as third, naked. Main flagellum of six segments, with one aestethascs and simple setae. Accessory flagellum of three segments, with simple and penicillate setae.

Antenna (Fig. 5B) peduncle first article as long as wide, inner lobe with two small setae. Second article 1.5 times as long as first, 1.5 times as long as wide, with inner medial small seta and outer subdistal seta, squama with long marginal setae. Third article one-third length of second, with long simple seta. Fourth (common) article about twice as long as third, with penicillate inner seta. Fifth article 1.3 times as long as two preceding articles together, with penicillate setae. Flagellum of six segments, first segment with long simple setae on outer margin, rest with simple and penicillate setae.

Mouthparts. Labrum (Fig. 5C) bilobed, setulose. Left mandible (Fig. 5D) with strongly dentate pars incisiva, lacinia mobilis tridentate; setiferous lobe with five bifurcate and trifurcate setae; pars molaris triturative. Right mandible (Fig. 5E) as left but without lacinia mobilis. Mandibular palp three-articled; first article about 1.5 times as long as wide, with numerous setae on inner margin; second article 1.3 times as long as first, with four pectinate short setae; distal article with several simple and pectinate setae. Maxillule (Fig. 5F) inner endite with marginal setae and apophysis and five distal pectinate setae; outer endite with ten distal blunt spines and two subdistal pectinate setae, outer margin with fine setae; palp with four long, minutely plumose setae. Maxilla (Fig. 5G) outer lobe of fixed endite with bifurcate, trifurcate and pectinate spines; inner lobe of fixed endite with a row of simple setae (partly broken); outer lobe of moveable endite with serrate setae. Labium (Fig. 5H) outer margin serrated, palp with three distal spines and marginal setules. Maxilliped (Fig. 5I) inner distal corner with spine; palp first article with outer distal short seta and inner long seta; second article inner margin with numerous setae, outer margin with one long distal seta; third article with setae along inner margin; distal article with distal setae; endite (Fig. 5I′) inner margin with four coupling-hooks, ventral margin with eight setulose spines, distal margin with row of simple and row of blunt setae, and two outer long setae. Epignath (Fig. 5J) with a long, distally setulose spine.

Cheliped slender (Figs. 6A, 6A′). Basis subtriangular, 1.9 times as long as wide, with mid-ventral stout spine, simple setae on dorsal margin and plumose setae on ventral margin. Exopodite present, first article cylindrical, naked, distal article with four or five plumose setae. Merus elongate, with medial and ventrodistal groups of setae. Carpus slender, 2.5 times as long as wide, with simple setae. Chela twice as long as wide; palm about as long as wide, with dorsal simple seta and tuft of simple dorsal setae near dactylus insertion on inner side (Fig. 6A′), and lateral simple setae on outer side (Fig. 6A); fixed finger with row of ventrodistal setae; cutting edge almost straight, with row of setae. Dactylus as long as fixed finger, with three medial setae, cutting edge with small spines.

Pereopod 1 (Fig. 6B) coxa with pronounced anterodistal apophysis, with simple setae. Basis twice as long as wide, with simple setae along anterior and posterior margins, and tuft of ventrodistal setae; exopodite as on cheliped. Ischium with simple ventrodistal setae. Merus narrower proximally, 1.8 times as long as wide, with simple setae, and ventrodistal spine. Carpus 0.5 times as long as merus, with marginal setae, two ventral spines and one dorsodistal spine. Propodus about as long as carpus, with marginal setae and two dorsodistal spines. Dactylus with ventral spiniform apophysis, two mid-dorsal simple seta, and distal setule. Unguis about one-quarter of length of dactylus.

Pereopod 2 (Fig. 6C) coxa with setae. Basis 2.2 times as long as wide, with dorsal and ventral simple setae, and tuft of ventrodistal setae. Ischium with tuft of ventrodistal setae. Merus, carpus and propodus with long marginal setae. Merus narrower at base, 1.4 times as long as wide, with one ventrodistal, long spine. Carpus about as long as merus, with dorsodistal serrate spine. Propodus 2.7 times as long as wide, with three serrate spines. Dactylus 0.6 times as long as propodus, with distal setule, unguis about one-quarter of length of dactylus.

Pereopod 3 (Fig. 6D) coxa with setae and oostegite. Basis 2.4 times as long as wide, with simple and penicillate setae and tuft of ventrodistal setae. Ischium with ventrodistal setae. Merus, carpus and propodus with long ventral and lateral setae. Merus narrower at base, about as long as wide, with one ventrodistal serrate spine. Carpus 1.4 times as long as wide, with long ventral spine. Propodus 2.5 times as long as wide and about as long as carpus, with one dorsal penicillate seta, two ventral and three dorsal serrate spines. Dactylus as long as propodus, with one short seta. Unguis about one-third the length of dactylus.

Pereopod 4 (Fig. 6E) basis twice as long as wide, with penicillate setae and two ventrodistal setae. Ischium with ventrodistal tuft of setae. Merus, carpus and propodus with long ventral setae. Merus with one pair of ventral serrate spines and dorsodistal short seta. Carpus about twice as long as merus, with row of four ventral serrate spines becoming longer approaching propodus. Propodus 2.2 times as long as wide and 0.8 times as long as carpus, with one dorsal penicillate seta, row of long pectinate terminal setae. Dactylus about as long as propodus, with one fine short setule near unguis. Unguis one-third of length of dactylus.

Pereopod 5 (Fig. 6F) basis thrice as long as wide, with two medial penicillate setae and two ventrodistal simple setae. Ischium with ventral setae. Ischium, merus, carpus and propodus with ventral setae. Merus with one ventral spine. Carpus twice as long as merus, with two rows of three spines, becoming longer approaching propodus. Propodus 1.7 times as long as wide and 0.8 times as long as carpus, with one dorsal penicillate seta, two pairs of ventral serrate spines, and row of terminal pectinate setae. Dactylus about as long as propodus, with distal seta. Unguis one-quarter of length of dactylus.

Pereopod 6 (Fig. 6G) basis oval, twice as long as wide, with row of dorsal and ventral plumose setae. Ischium with a ventrodistal tuft of simple setae. Merus and carpus with ventral simple setae and dorsal long plumose setae. Merus narrower proximally, with one serrate spine. Carpus with three ventral spines, longer towards propodus, and medial spine. Propodus ovate, 0.8 times as long as carpus, with row of pinnate spines and two longer, serrate spines along ventral and terminal border, one medial spine and two simple dorsal setae. Dactylus about as long as propodus, with one medial and one distal setules. Unguis about one-third of length of dactylus.

Pleopod 1 (Fig. 6H) basis with three outer and four inner plumose setae, pleopod 2 with four outer and four inner plumose setae, pleopod 4 with three outer and three inner plumose setae, pleopods 3 and 5 incomplete. Both rami with numerous distal and outer marginal plumose setae, endopod with two inner marginal setae.

Uropod (Fig. 6I) protopod with three setae. Endopod with 25 to 30 segments, some with simple or penicillate setae. Exopod three-segmented; distal segment significantly longer than the other two together, with three distal setae.

Distribution: Species known only from the type locality

Remarks

The females of this species are most similar to Apseudopsis acutifrons, which was partially illustrated by Sars in a subsequent publication (Sars, 1886) and later on by Guţu (2002). Both species share a combination of four ventral spines on pereopod 1 and apophyses directed forward on pereonites 3–6, but A. salinus sp. nov. differs in lacking posterolateral apophyses on pereonite 6, having hyposphenia on pereonites 1–6 (only on pereonites 2, 3 5 and 6 in A. acutifrons, as seen in Guţu, 2002), and a much less acute rostrum (very straight and narrow in A. acutifrons, according to the illustrations by Sars (1886)).

Identification key for the Atlantic and Mediterranean Apseudopsis species

1. Apophysis on pleonite 5 twice as long as apophyses on remaining pleonites A. larnacensis sp. nov.

- Apophyses on pleonites equal in length2

2. Pereopod 3 basis with posterior projection proximally A. daria Esquete & Tato, 2024

- Pereopod 3 basis without posterior projection proximally3

3. Pereopod 1 merus with dorsodistal spine 4

- Pereopod 1 merus without dorsodistal spine 5

4. Pereonites 2–6 with apophyses on anterior and posterior cornersA. elisae (Băcescu, 1961)

- Pereonites without anterolateral or posterolateral apophysesA. latreillii (Milne Edwards, 1828)

5. Pereonites without apophyses A. rogi Esquete, 2016

- Some pereonites with anterolateral or posterolateral apophyses or pointed corners 6

6. Pereopod 1 propodus with five or more ventral spines 7

- Pereopod 1 propodus with four or fewer ventral spines11

7. Pereopod 1 propodus with five ventral spines 8

- Pereopod 1 propodus with six ventral spines10

8. Rostrum short9

- Rostrum relatively long, acute, with rounded shoulders A. cuanzanus Bochert, 2012

9. Pereonite 1 with posterolateral acute apophyses A. annabensis (Guţu, 2002)

- Pereonite 1 without posterolateral acute apophysesA. formosus Carvalho, Pereira & Esquete, 2019

10. Rostrum short, pointed; all pereonites with posterolateral apophyses; female with hyposphenia on pereonites 2 and 6; male chela fixed finger with semicircular apophysis A. arguinensis (Guţu, 2002)

- Rostrum long, acute; pereonites 2–5 with short posterolateral hooks; females with hyposphenia on pereonites 2–6; cheliped not sexually dimorphic, slender, merus about 3.5 times as long as broad, chela cutting edges not ornamented A. isochelatus Guţu, 2006

11. Pereopod 1 propodus with three ventral spines12

- Pereopod 1 propodus with four ventral spines14

12. Pereonites 3 and 4 clearly longer than remaining pereonites A. minimus (Guţu, 2002)

- Pereonites 3–4 similar in length to pereonites 2 and 513

13. In females hyposphenium present on pereonite 3, antennule main flagellum with 8 to 10 segments A. apocryphus (Guţu, 2002)

- Hyposphenium absent on pereonite 3, antennule main flagellum with 6–7 segments A. gabesi Esquete, 2019

14. At least some pereonites with anterolateral apophyses 15

- No pereonite with anterolateral apophyses17

15. Anterolateral apophyses on pereonites 3–6 directed forward 16

- Anterolateral apophyses on pereonites 3–6 not directed forward, but perpendicular to the main axis of the body A. ostroumovi (Băcescu & Carausu, 1947)

16. Pereonite 6 with posterolateral apophyses A. acutifrons (Sars, 1882)

- Pereonite 6 without posterolateral apophyses A. salinus Esquete, Jóźwiak, Stępień, sp. nov.

17. Pereonites 1–6 with posterolateral apophyses; mature specimens’ accessory flagellum of antennule of 6–8 segments 18

- Only pereonites 2–6 with posterolateral apophyses; mature specimens’ accessory flagellum of antennule of ten segments; females with hyposphenium on pereonite 6 only; male chela fixed finger with semicircular and triangular apophyses on cutting edge A. adami Esquete & Bamber, 2012

18. Females with hyposphenia on pereonites 2, 3 and 6. Male chela with apophysis on fixed finger cutting edge and proximal apo-physis on dactylus cutting edge A. bacescui (Guţu, 2002)

- Females with hyposphenia on pereonites 2 and 6. Male cheliped fixed finger cutting edge with proximal semicircular apophyses and invagination A. mediterraneus (Guţu, 2002)

Bathymetric and habitat preferences of representatives of the genus Apseudopsis

Within 24 studied species of Apseudopsis, 22 have been collected from shallow waters, not exceeding 200 m in depth and within a relatively narrow depth range (Table 1). Three species are exceptions: A. latreillii was found from depths ranging from 0 to 356 m, and A. uncidigitatus from 13 to 800 m, while A. isochelatus was recorded from a narrow depth range but at depths around 1,000 m.

Table 1 Depth and habitat preferences of Apseudopsis species.

Species	Depth (m)	Habitat	Source	
Min	Max	
Apseudopsis acutifrons	10	200	Fine mud, soft substrata including all phanerogames	Bamber et al. (2009), Bakir et al. (2014, 2024), Lubinevsky, Tom & Bird (2022)	
Apseudopsis adami	0.3	7	Zostera meadow; sand, gravelly sand, muddy sand	Esquete et al. (2012a), Carvalho et al. (2019), Garcia Herrero et al. (2021)	
Apseudopsis annabensis	0.8	11	Fine sand	Esquete, Fersi & Dauvin (2019)	
Apseudopsis apocryphus	6.6	120	Fine mud	Bamber et al. (2009), Lubinevsky, Tom & Bird (2022)	
Apseudopsis bacescui	13	13	Sandy mud	Guţu (2002), Carvalho et al. (2019)	
Apseudopsis bruneinigma	60	60	Fine sand and silt	Bamber (1999)	
Apseudopsis caribbeanus	12	12	Sand and mud	Guţu (2006)	
Apseudopsis cuanzanus	46	46	No data	Bochert (2012)	
Apseudopsis daria	5.2	12.9	Claye mud, mud with shells	Senckenberg Ocean Species Alliance et al. (2024)	
Apseudopsis elisae	5	97; 122*	Sand	Băcescu (1961), *Lubinevsky, Tom & Bird (2022) (not certain)	
Apseudopsis formosus	2.5	11	Muddy sand	Carvalho et al. (2019)	
Apseudopsis gabesi	0.8	11.9	Fine, medium or silty sand	Esquete, Fersi & Dauvin (2019)	
Apseudopsis hastifrons	39	39	No data	Guţu (2002)	
Apseudopsis isochelatus	1,000	1,000	No data	Guţu (2006)	
Apseudopsis larnacensis sp. nov.	11	11	Muddy sand		
Apseudopsis latreillii	0	356	Meadows of Zostera marina and Z. noltii; among Caulerpa prolifera, Posidonia oceanica; soft substrata including all phanerogames; hard substrata including sponge and mussels	Cacabelos, Gestoso & Troncoso (2009), Chicharo et al. (2002), Guerra-García & García-Gómez (2004), Fišer (2004), Marín-Guirao et al. (2005), Sanz-Lázaro & Marín (2006), Sánchez-Moyano, Garcia-Asencio & Garcia-Gómez (2007), Lourido, Moreira & Troncoso (2008), Moreira, Lourido & Troncoso (2008), Varela, Moreira & Urgorri (2009), Bamber (2011), Esquete et al. (2012b), Bakir et al. (2014, 2024), Garcia Herrero et al. (2021)	
Apseudopsis mediterraneus	0.8	41/138*	Fine and medium sand; soft substratum including all phanerogames	Bamber et al. (2009), Bakir et al. (2014, 2024), Carvalho et al. (2019), Esquete, Fersi & Dauvin (2019), Lubinevsky, Tom & Bird (2022)	
Apseudopsis minimus	8.9	54	Hard substratum including algae, sponge, mussels	Bamber et al. (2009), Bakir et al. (2014, 2024), Lubinevsky, Tom & Bird (2022)	
Apseudopsis olimpiae	12	120	No data	Hansknecht & Heard (2001)	
Apseudopsis opisthoscolops	2	11	Sand, mud flat and algae covering the pneumatophores of Avicennia marina	Bamber, Chatterjee & Marshall (2012), Bamber (2013)	
Apseudopsis ostroumovi	0	129	Fine and medium sand	Băcescu & Carausu (1947), Guţu (2002), Bamber et al. (2009), Esquete, Fersi & Dauvin (2019)	
Apseudopsis rogi	6	11	Sand	Esquete, Ramos & Riera (2016)	
Apseudopsis salinus sp. nov.	11	11	Muddy sand		
Apseudopsis tuski	10	84	No data	Błażewicz-Paszkowycz & Bamber (2007)	
Apseudopsis uncidigitatus	13	816	Sand	Carvalho et al. (2019), Garcia Herrero et al. (2021)	
Note:

* Most probably incorrect according to Bamber et al. (2009).

Species of the genus Apseudopsis were typically recorded from sandy or muddy substrates (Table 1). Additionally, some specimens of A. latreillii have been collected in association with the alga Caulerpa prolifera or among Posidonia oceanica meadows, while A. adami has been found in association with Zostera meadows (Table 1).

Distribution of representatives of the genus Apseudopsis

Species of Apseudopsis are distributed between 60°N and 40°S. The most northern species currently known is Apseudes latreillii noted from the vicinity of Norway, while the southernmost species is A. tuski noted from the Bass Strait. Fifteen species are found in the Mediterranean (including the two species described herein), eight along the Atlantic coasts of Europe, three from Atlantic coast of Africa and two each from Indonesia, the Black Sea, and the Gulf of Mexico, and one each from Tenerife (Canary Island), the Red Sea, and Australia, respectively.

Representatives of the genera Apseudopsis are noted from 25 ecoregions. The ecoregion with the highest number of species is the Levantine Sea, where ten species are known (with two newly added due to current investigation (Table 2)). The second most rich in species is the South European Atlantic Shelf, with seven recognized species. Six species are recorded from the Western Mediterranean, which makes it rank third in terms of diversity (Table 2).

Table 2 Number of Apseudopsis species, records and endemic species calculated for ecoregions.

Ecoregion	Richness	Records	Endemic species	
Adriatic Sea	2	7	1	
Aegean Sea	4	36	0	
Alboran Sea	2	18	0	
Angolan	1	1	1	
Azores Canaries Madeira	2	8	1	
Bassian	1	20	0	
Black Sea	3	151	0	
Cape Howe	1	38	0	
Carolinian	1	84	0	
Celtic Seas	2	514	0	
Floridian	1	2	0	
Greater Antilles	1	1	1	
Ionian Sea	2	4	0	
Levantine Sea	10	129	5	
North Sea	1	83	0	
Northern and Central Red Sea	1	1	1	
Northern Gulf of Mexico	1	27	0	
Palawan/North Borneo	1	7	1	
Saharan Upwelling	1	2	0	
Sahelian Upwelling	2	2	1	
South European Atlantic Shelf	7	280	2	
Southern Norway	1	7	0	
Tunisian Plateau/Gulf of Sidra	4	20	1	
Western Bassian	1	2	0	
Western Mediterranean	6	344	0	

Apseudopsis latreillii is the species with the widest geographical distribution, as it is present in 12 ecoregions. Among the most widespread species are also A. mediterraneus and A. acutifrons, both recorded from five ecoregions. A. latreillii is also the most frequently recorded species, with 1,267 distinct records, followed by A. mediterraneus with 137 distinct records and by A. ostroumovi with 125 records.

Fifteen species are characterized by a narrow distribution, found from only one ecoregion (Table 2). Five of them are known only from their type locality, and one record only (A. caribbeanus, A. cuanzanus, A. erythraeicus, A. isochelatus, and A. tridens).

To the ecoregions with the highest number of records belong the Celtic Sea with 514 records, the Western Mediterranean with 344 records and the South European Atlantic Shelf with 280 records (Table 2), respectively.

Discussion

Bathymetric and habitat preferences of representatives of the genus Apseudopsis

Representatives of Apseudopsis predominantly inhabit shallow waters, a characteristic common among other Tanaidacea, which are recognized as members of a phylogenetically old group (Sieg, 1983; Błażewicz-Paszkowycz & Bamber, 2012). Although two species have been recorded beyond the continental shelf, the limited number of deep–sea records is insufficient to explain their unusual depth preferences or discuss their adaptation to life on the slope. The low number of scientists working on deep-sea Tanaidacea suggests that our knowledge in this area is unlikely to expand soon.

The Apseudopsis preference for muddy and sandy substrates is consistent with the morphological adaptations of these species, characterized by a large and broad first pair of pereopods with a shortened dactylus. It is suggested that this type of pereopod morphology (common for other members of the family Apseudidae as well as families like Parapseudidae, Sphyrapodidae or Whiteleggiidae) is an adaptation for swimming or burrowing in sand or mud (Larsen, 2005). Indeed, specimens of A. latreillii and A. adami have been observed using the first pair of pereopods for digging in the sediment after collection during sampling (P Esquete, 2012, personal observation).

Although the first pereopod is too robust and heavily armed to be suitable for tube construction—a behavior typical of the majority of representatives of the suborder Tanaidomorpha (Larsen, 2005)—some individuals of Apseudopsis have been found inhabiting tubes. These tubes, likely secreted by other organisms, may offer protection from predators and environmental stressors, particularly in less stable or exposed habitats. This behavior complements their burrowing tendencies and underscores their ability to exploit a range of microhabitats.

Distribution of representatives of the genus Apseudopsis

The greatest species richness of Apseudopsis is observed in the eastern part of the Mediterranean Sea, particularly in the Levantine Sea. Long-term studies near the Israeli coast have identified eight species of Apseudopsis (Bamber et al., 2009; Guţu, 2002; Lubinevsky, Tom & Bird, 2022). Our study has revealed two additional species from the vicinity of Cyprus, reinforcing the Levantine Sea’s status as the most species-rich area for Apseudopsis. Five Levantine species are widespread taxa and are found also in the other part of the Mediterranean as well as in the Black Sea (A. acutifrons, A. ostroumovi, A. elisae, A. mediterraneus) or in the Atlantic (A. latreillii). The remaining species are, so far, recognized only from the ecoregion. This species composition underscores the distinctiveness of the Levantine ecoregion. It is suggested that the fauna here contains tropical and subtropical elements (survivors of the Messinian salinity crisis—a period of complete disconnection from the Atlantic—and Pleistocene interglacial). To a lesser degree the Levantine Sea has been influenced by warm-temperate Atlantic-Mediterranean fauna (Por, 1989; Bianchi et al., 2012). Currently the basin is influenced by significant influxes from the Red Sea (Bianchi & Morri, 2000; Coll et al., 2010; Galil et al., 2017). It remains uncertain whether the diversity of Apseudopsis in the Levantine Sea is affected by this invasion, due to the limited information available about benthic peracarids in the Red Sea. So far, only one Apseudopsis species—A. erythraeicus from the Red Sea, not recorded in the Levantine Sea, is known.

The second most species-rich ecoregion is the South European Atlantic Shelf. Knowledge about the diversity in this region primarily stems from investigations in the Ria Formosa lagoon and the Algarve coast (Carvalho et al., 2019). It is suggested that the diversity of Apseudopsis in this area results from the intersection of fauna from the Mediterranean (A. bacescui, A. mediterraneus, A. elisae, A. uncidigitatus) with that from the northern Atlantic coast of the Iberian Peninsula (A. adami, A. latreillii). Species would have been transported to the lagoon via water currents and due to intense sedimentary transport caused by strong waves in the Gibraltar Strait (Carvalho et al., 2019). Due to the variety of habitats, including salt marshes, mud flats or sand newcomers found shelter (Newton & Mudge, 2003) and survived in new area. So far, one endemic species is known from the lagoon (A. formosa). The recent discovery of the species A. daria Esquete & Tato, 2024, in the northern part of the South European Atlantic Shelf, increases the number of species known from this ecoregion and highlights the poorly recognized diversity of temperate members of the genus.

Only two species of Apseudopsis have been recorded in the Alboran Sea, a region suggested to be a biogeographic pathway connecting the Atlantic and Mediterranean (Real et al., 2021). Given the threefold higher diversity in neighboring ecoregions, the actual diversity in the Alboran Sea is likely much higher, underscoring that some parts of the Mediterranean remain undersampled.

Our investigation revealed a significant disparity in the number of records across different ecoregions. For instance, the number of records from the Western Mediterranean and the Celtic Sea is several dozen times higher compared to other regions. Most of these records originate from extensive monitoring programs. Data for the Celtic Sea was provided by the Regional Seabed Monitoring Plan (RSMP), which collected samples along the UK coast over a 48-year period from 1969 to 2016 (Cooper & Barry, 2017). Additional data came from the UK Marine Recorder public snapshot. Distribution information for the Western Mediterranean is derived from the Coastal Monitoring Database and related products collected since 1974, as well as the French Mediterranean Lagoon Monitoring Network (RSL project), operational since 2006 (gbif.org). Notably, 95% of the records were attributed to a single species—A. latreillii (Fig. 7). However, given the high number of sympatric species identified in detailed examinations of Apseudopsis collections from the Algarve coast (Carvalho et al., 2019) and the Gulf of Gabes (Esquete, Fersi & Dauvin, 2019), and the fact that A. latreillii may co-occur with other species (Esquete et al., 2012b; Esquete, Fersi & Dauvin, 2019), it is possible that the A. latreillii recorded in these monitoring programs represents several species.

Figure 7 Distribution of Apseudes latreillii, based on two sources of information: online databases and literature.

The limited number of species from regions outside the Mediterranean and adjacent Atlantic waters, combined with the scattered data (two species from Indonesia, two from the Gulf of Mexico, one from temperate Australia) hinders a comprehensive understanding of the factors shaping their worldwide distribution. The gaps in species records from other areas make it difficult to determine whether this distribution is the result of undersampling or reflects actual biodiversity patterns.

Conclusions

Two new species of Apseudopsis have been described from the vicinity of Larnaca, Cyprus, increasing the total number of Apseudopsis species recorded in the Mediterranean Sea to fifteen. This makes the Mediterranean the most diverse region for the genus. As a result, the area remains a focal point for biodiversity research, revealing patterns of species distribution that are not as evident in other parts of the world. Species composition, with Atlantic–Mediterranean, Black Sea–Mediterranean and endemic species reflects the complex history of the sea, while preferences of shallow waters and sandy and muddy substrates, suggest that these environments play a crucial role in maintaining Apseudopsis populations.

Based on current knowledge, the Mediterranean Sea can be considered a potential center of speciation for Apseudopsis, although a more comprehensive understanding of global distribution patterns requires further data.

Our study also underscores the importance of involving taxonomic specialists in biodiversity monitoring programs. Careful examination of specimens is critical for accurately assessing the true diversity of Apseudopsis, as demonstrated by recent investigations of collections from the Gulf of Gabes and the Ria Formosa lagoon.

Supplemental Information

Supplemental Information 1 Raw distribution data of the genus Apseudopsis.

The coordinates, depth and year of collection collected for all Apseudopsis representatives.

The authors greatly appreciate the invaluable help of Vasilis Resaikos, and Magdalene Papatheodoulou (both Enalia Physis, Cyprus) with sampling and sample preparation and help of Pablo Saenz Arias with further lab work. We would like to thank the reviewers for their valuable comments, which helped improve this manuscript. We are grateful to Vasilis Louca (University of Aberdeen) and Dimitrios Xevgenos (TU Delft) for useful discussions.

Additional Information and Declarations

Competing Interests

Author Contributions

Data Availability

New Species Registration

Sergio C. Garcia Gómez is employed by Tecnoambiente S.L.

Anna Stępień conceived and designed the experiments, performed the experiments, analyzed the data, prepared figures and/or tables, authored or reviewed drafts of the article, and approved the final draft.

Piotr Jóźwiak conceived and designed the experiments, performed the experiments, analyzed the data, prepared figures and/or tables, authored or reviewed drafts of the article, and approved the final draft.

Sergio C. Garcia Gómez conceived and designed the experiments, authored or reviewed drafts of the article, and approved the final draft.

Eleni Avramidi conceived and designed the experiments, authored or reviewed drafts of the article, and approved the final draft.

Kleopatra Grammatiki conceived and designed the experiments, authored or reviewed drafts of the article, and approved the final draft.

Myrsini Lymperaki conceived and designed the experiments, authored or reviewed drafts of the article, and approved the final draft.

Frithjof C. Küpper conceived and designed the experiments, authored or reviewed drafts of the article, and approved the final draft.

Patricia Esquete conceived and designed the experiments, performed the experiments, analyzed the data, authored or reviewed drafts of the article, and approved the final draft.

The following information was supplied regarding data availability:

The distribution data are available in the Supplemental File.

The following information was supplied regarding the registration of a newly described species:

Publication LSID: urn:lsid:zoobank.org:pub:72BC0082-9C17-4901-8D9A-547B073760EB.

Apseudopsis larnacensis apseudopsis larnacensis LSID: urn:lsid:zoobank.org:act:29DDF9A0-39DC-44C9-90EB-DD90EACAD287

Apseudopsis salinus apseudopsis salinus LSID: urn:lsid:zoobank.org:act:BBB0B29E-8181-4714-9FD5-E79DFFE5187C.

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
