# Peer review of "Description of two new Apseudopsis species (A. larnacensis sp. nov and A. salinus sp. nov.) (Tanaidacea: Crustacea) from the Mediterranean and a biogeographic overview of the genus"

_PeerJ, doi:10.7717/peerj.18740_

## Round 0.1 · original submission · Major Revisions

Dear Dr. Stepien

The reviewers have commented on your manuscript. You can find the attached reports. Based on the comments and suggestions of the expert reviewers, a major revision is needed for your article.

I request you check and correct the manuscript based on the reports.

Sincerely

Reviewer 1 ·

Basic reporting

.

Experimental design

.

Validity of the findings

.

Additional comments

This is well prepared manuscript, providing useful informations on distribution and morpohology, accomapined with a high quality illustrations and photos, for two new species of the genus Apseudopsis collected from Larnaca, Cyprus. This paper merits publication in PeerJ.

Reviewer 2 ·

Basic reporting

The text is generally well-presented but appears not have been spell-checked. There are various places where the English could be improved or changed (see annotated Word file returned).

The actual context of what is basically a description of two new species seems overblown, especially the Introduction. More reference solely to previous publications would have sufficed in many instances.

The figures are very good but there appears to be at least two instances (Figs 2 & 7) where a mix-up between items has been made.

There is too much repetition of species authorities after the first use.

Some sections of the descriptive text could be refined - or at least are ambiguous now. (see annotated Word file).

Experimental design

My main criticism is that the authors have not fully examined the available data for certain species, namely those from the Israeli coast. They could easily interrogate the IOLR (Haifa) database for hundreds of records of Apseudopsis, or even reach out to G J Bird who was one of the principal authors in Lubinevsky et al 2022. They seem not to have even read this manuscript.

I am also not sure just how far they have progressed the findings about the distribution of the genus, considering that Carvalho et al 2019 did a very similar study and produced figures (maps) of far greater quality and attractiveness.

Validity of the findings

The essential novelty of this manuscript is the bare description of two new species, one of which (salinus) might just be a variant or part of A. acutifrons.

Rather than repeat essentially the same work of Carvalho et al, it would have been more useful for future workers on the genus (especially in the Mediterranean) or those identifying material for ecological surveys, for the authors (all eight of them!) to have produced an updated key to the species. This is what this type of paper are really for.

Additional comments

Other comments and suggested changes to the text can be seen in the returned Word file.

Apologies for the apparent formatting mess in places - it was converted from a PDF file as I do not have Acrobat for reviewing/editing.

Annotated reviews are not available for download in order to protect the identity of reviewers who chose to remain anonymous.

Reviewer 3 ·

Basic reporting

The manuscript (MS) indeed had a priority to describe and report two new anisopod (Tanaidacea) species. Description was performed well by the co-authors experts on the tanaids. However, some literature were missing for the distribution of the genus Apseudopsis as follows to contribute the MS:
1. BAKIR, AHMET KEREM; KATAĞAN, TUNCER; AKER, HALİM VEDAT; ÖZCAN, TAHİR; SEZGİN, MURAT; ATEŞ, ABDULLAH SUAT; KOÇAK, CENGİZ; and KIRKIM, FEVZİ (2014) "The marine arthropods of Turkey," Turkish Journal of Zoology: Vol. 38: No. 6, Article 6. https://doi.org/10.3906/zoo-1405-48
2. AHMET KEREM BAKIR, HALİM VEDAT AKER, ÖZGE ÖZGEN, FURKAN DURUCAN 2024. Diversity of marine Arthropoda along the coasts of Türkiye. Turk J Zool, (2024) 48: 446-530

The MS could be updated using the former and recent publications above.
In general, the crustaceans are very flexible and fragile in the description of the new species or not.
Some crustacean especially copepods, cumaceans, tanaids could have some slight difference whether the specimens could be new or not.
Some tanaids could be simultaneous hermaphrodite and have sexual dimosphism in the tanaids. All specimens are females in the MS. Hyposphenia could be negliable in some cases particularly for preparatory females.
Why did the authors use genetic analyses to define and enhance the specimens as new species. Nowadays, to ensure such description the genetic method could be applied often besides the morphological description.
Some points above need to be justified and clarified.

Experimental design

Material examined was only specimen of holotype and paratype are missing to show variety on the morphological description. Sex was only female. Number of examine materials could not be sufficient for the description. The MS seems to have one material for each new species. Many description published before included the paratype and sex. The MS needs to detail such materials.

Validity of the findings

The MS needs to enhance validification of the finding regarding to the comments aforementioned before the acceptance of the MS.

Additional comments

Remarks could be shown on the figures with arrows.

·

Basic reporting

1. BASIC REPORTING
I am not a native speaker so I can only state if the language is good enough to understand the topic.

The paper is well written but has a lot of mistakes and discordance between text and figures specially figure figure 2
The use of databases is very nice but must be taken with a lot of care specially for not well studied taxa as Tanaidacea. My personal experience is that about 80% of the reported fauna is misidentified. Only reports made by recognized taxonomists can be take seriously. This must be taken into account when using the databases I know it is not nice but that is what we have. There are a lot of fairytales in the databases.

The paper at it actual state can not be accepted for publication it need carefully examination of the text with the figures and one species support is very weak it can be simple an ecophene of other species.

Experimental design

It is a descriptive paper so there is no experimental design, the methodology used is adequate for this kind of research.
The description of the setation is very poor and in many cases not enough. Setation position and number is important at species level, to support and justify the identification of a species. This must be improved.

Validity of the findings

The first species is problematic Apseudopsis larnacensis in the remarks it is stated that is almost identical to Apseudospsis hastifrons and only one character is the difference this can be an ecophene. There must be more strong arguments for the erection of the species.

The use of the databases data is still problematic, is a problem with the databases origin of data. Too few taxonomic expertise is flowing in the data.
Many things are still part of the realm of fairy tales

Additional comments

4. General comments
The paper has still a lot of problems in description and wording.


Line 63 The complex biodiversity was shaped by the geological events in the Tertiary period and significant climatic changes in the Quaternary.
The Mediterranean was dry for almost 800.000 years about 5 million years ago and the salt flats were a dead trap to everything only with the filling of the basin the marine life returned to the sea. All the fauna and flora of the Mediterranean is new, that is may be the main reason for the high diversity (other than is one of the most studied seas in the world).

Line 86 …difficult due to small sizes,…. 90 % of tanaidacea are less than 5 mm so it is part of the job not a difficulty. The animals described are more than 5 mm also are giants in the Tanaidacea world.

Line 241why is there no diagnosis of the genus or a reference to the latest diagnosis done by somebody?

Line 245 museum number is missing.
Line 248 give the name of the city, is important to prove the declination of the name.
Line 261 …margins of all pleonites produced posteriorly,… what do you mean what do they produce? the animals are already death so can not produce anything.
Line 262 …Pelonite 5 slightly longer, projections visibly larger….what is a pelonite, which projections are larger assuming the pleonites have projectiosn how can the reader know?
Line 275 how many setae on squama? Describe!
Line 276 …short, naked…Be consistent if all other articles are defined by length and wide why is the third only short? What is short? In reference to what?

Line 277 ..Flagellum of six segments,….why are 7 segment on the drawing? Something is flawed. The setules are not only simple there are very long and short ones. Describe properly.

Line 281 .. why is the labrum not figured? With 10 animals why was only one dissected? No variability expected?

Line 282 the figure 2C and 2D must be misplaces the tridentated lacina mobilis is on figure 2D not on 2C the distal trifurcate spine is also on figure 2C. check carefully.

Line 288 Only 4 pectinate setae on drawing and one simple seta.

Line 298. The endite has the coupling hooks on one side and the setulose setae on the other side not all on the same side. The whole description of the maxilliped is very confusing and must be redone carefully, dummies proof.

Line 301 cheliped has a coxa or a direct insertion?
The description of the cheliped is horrible. There is a clear distal projection on the merus, not mentioned, in the carpus there are several rows of obiously sensorial setae and and are treated as normal simple setae. The chela is the inner view the same as the outher view? Are ther sensory setae on the inner view like in most Tanaidaceans or not?

Line 316 Ungius sharp blunt or something else?

Line 367 if the only difference with Apseudopsis hastifrons is very weak as you state in the remarks it can be a simple ecophene for the area.

Line 381 the etymology is horrible if it is derived of the brine state that. It is not a geographical reason a place. Explain better.

Line 401 how many setae?

431 the epignath has some sort of structure opposite the spine what is that why it is not mentioned. May be important for species definition.

Line 433 cheliped insertion simple or via coxa?
Line 437 inner view or outher view of chela? Setae position not described.
Line 457 why is there no drawing of the oostegite?
Line 495 Position of the setae is important at species level why it is not addressed?

Several findings of the distribution of the genus can be better explained by the number of tanaidaceologist working with material of the area than of real distribution. Deep sea distribution of tanaidacea is correlated to tanaidaceologists works. But to justify the existence of the databases is ok.

Reviewer 5 ·

Basic reporting

English needs to be checked by a native speaker.

Experimental design

no comment

Validity of the findings

no comment

Additional comments

This study is interesting, with new information on Apseudopsis genus. This study also compiles the geographical distribution of species of this genus in different regions. I think it will provide an important contribution to literature. Overall, the manuscript was well prepared. All sections were explained clearly and comprehensively. All figures and tables were well prepared and necessary.

Suggested minor corrections;
Line 41: Pollution is not a kind of human activity. ‘the results of human activities’ should be written. Or ‘anthropogenic effects’ should be preferred instead of ‘human activities’.
Lines 45-47: Please check the grammar (Sequence of tenses)
Line 101: Please write ‘’Lang’s’’ instead of ‘’his’’.
Lines 150-153: Please check the grammar
Lines 178: samples or sediments?
Line 245: museum number?
Line 378: museum number?
Lines 511-515: please add references.
Line 565 and line 581: If you refer to Figures 7 and 8, you will make it easier for readers.
Line 580: please provide more information (species name)
Line 594: Please delete ‘interestingly’. Actually, this is an expected situation. Due to tanaid crustaceans from the Alboran Sea have been poorly studied, only two species from this genus may have been reported.

---

## Round 0.2 · Minor Revisions

Dear Dr. Stepien

The reviewers have commented on your manuscript. You can find the attached reports. Based on the comments and suggestions of the expert reviewers, a minor revision is needed for your article.

I request you check and correct the manuscript based on the reports.

Sincerely

Reviewer 2 ·

Basic reporting

This manuscript seem to have been reasonably well revised but numerous small errors remain as well as a few concerning the morphology-taxonomy.

The main critique of the morphological analyses, which is also relevant to the key, is the description or mention of the pereonal apophyses, particularly the anterolateral pair. The key suggests that A. apocryphus lacks apophyses but these are actually quite prominent (Fig. 7 in Lubinevsky et al 2022), although not as extreme as in A. minimus (also Fig.7 and the capture attached here – of the holotype). I think this aspect could still be made more explicit.

The key has a fault in that it does not lead to a couplet “5”.

The spelling of latreillii is often wrong in several place, including caption for FIG.7.

There are several other minor changes or corrections that I’ve made in the text – some grammatical, others for style.

Thank you for asking me to review the paper again.
s

Experimental design

As for original review.

Validity of the findings

As for original review.

Additional comments

NO comments apart from that entered in the basic reporting section.
But see attached files.

Annotated reviews are not available for download in order to protect the identity of reviewers who chose to remain anonymous.

Reviewer 3 ·

Basic reporting

After significant number of all reviewers comments, the paper was well improved. The authors could update Fig. 7 for occurrence of the species along the Turkish Mediterranean Sea searching more papers. The paper could be accepted after the update.

Experimental design

The revised version is better than the previous version of the paper.

Validity of the findings

The paper is ready for acceptance even if there were no getenic analysis.

Additional comments

After significant number of all reviewers comments, the paper was well improved. The authors could update Fig. 7 for occurrence of the species along the Turkish Mediterranean Sea searching more papers. The paper could be accepted after the update.

Reviewer 5 ·

Basic reporting

The revised version looks good. The grammar of the manuscript has been improved. The MS includes sufficient introduction and background. All figures and tables have been well prepared and related to content.

Experimental design

Materials and methods are well explained.

Validity of the findings

Findings are well stated, linked to aims of the study.

Additional comments

-

---

## Round 0.3 · accepted · Accept

Dear Dr. Stepien,

I thank you for making the corrections and changes requested by the reviewers. I read and checked your valuable article carefully and am happy to inform you that the article has been accepted for publication in PeerJ.

Sincerely yours